# Precision public health: A natural experiment on chronic high-contrast PM$_{2.5}$ exposure and pulmonary function among older adults

**Hari Krismanuel** [ORCID]*, **Purnamawati Tjhin**

Faculty of Medicine, Universitas Trisakti, Jakarta, Indonesia

* hari_krismanuel@trisakti.ac.id

## Abstract

**Background:** While **fine particulate matter** (**PM$_{2.5}$**) is a respiratory hazard, most studies lack the temporal depth to distinguish chronic from acute effects, particularly among vulnerable older adults. This **natural experiment study employs with a 10-year stable residency filter.** To our knowledge, it is among the first in Southeast Asia to evaluate the decadal impact of PM$_{2.5}$ exposure on highly selected healthy older adults. Utilizing a Precision Public Health (PPH) framework, we provide **regional evidence in Southeast Asia** to integrate a **quasi-longitudinal approach** for isolating chronic respiratory effects using **standardized lung function** (**GLI 2012 – z-scores, adjusted for Southeast Asian populations**). **Methods:** We conducted a natural experiment involving 101 non-smoking older adults (65–80 years; normal BMI) in Indonesia. This design compared populations with **≥10-year residency** in **contrasting environments**: high-exposure urban (Kedoya) and low-exposure rural (Pangalengan). The decadal filter ensured **temporal precedence**. Spirometry included internal **BTPS (Body Temperature, Ambient Pressure, Saturated with water vapor)** compensation. **Multivariable linear regression** evaluated the association between exposure and Z-scores, adjusting for demographic and lifestyle covariates. **Results: High-exposure participants had significantly lower FEV$_1$_z and FVC_z than the low-exposure group, with no significant difference in FEV$_1$/FVC_z. Multivariable regression confirmed exposure group was the only independent predictor for FEV$_1$_z (β = −1.42, p < 0.001) and FVC_z (β = −1.14, p < 0.001), after adjusting for covariates. These findings indicate a subclinical reduction in lung volumes consistent with a non-obstructive, restrictive spirometric pattern.** Diagnostic testing indicated no violation of model assumptions was detected. **Conclusions:** High decadal PM$_{2.5}$ exposure is associated with significant standardized lung volume reductions. The 10-year residency stability criterion **enhances causal inference**. These findings advocate PPH approach, highlight the importance of z-score–based spirometric screening for early detection of **subclinical pollution-related lung function decline.** Integrating environmental risk assessment

**Data availability statement:** All relevant data are within the manuscript and its Supporting information files.

**Funding:** The author(s) received no specific funding for this work.

**Competing interests:** The authors have declared that no competing interests exist.

into geriatric care and air quality management offers a cost-effective pathway to mitigate long-term healthcare burdens in megacities.

## Introduction

Ambient air pollution, particularly fine particulate matter ($PM_{2.5}$), remains one of the leading environmental risk factors for morbidity and mortality worldwide [1–5]. $PM_{2.5}$ can penetrate deep into the distal airways, triggering inflammation, oxidative stress, and structural changes that contribute to impaired pulmonary function [1,3,6–9]. Older adults are especially vulnerable due to age-related physiological decline, higher prevalence of comorbidities, and reduced capacity to respond to environmental stressors [7,10,11]. **This heightened susceptibility underscores the need for a PPH approach, which moves beyond population-wide averages to focus on tailored promotive and preventive strategies for highly vulnerable subgroups** [12–14]. In this context, identifying individuals in the **'subclinical window'**—a **critical period** where physiological decline begins due to environmental stressors but has not yet manifested as overt disease—is essential for transitioning from reactive healthcare to proactive primary prevention.

Despite extensive evidence linking $PM_{2.5}$ exposure with adverse respiratory outcomes, the majority of epidemiological research has focused on children or middle-aged adults [4,8]. Studies involving older adults—who bear a disproportionate burden of air-pollution-related disease—remain relatively scarce, particularly in low- and middle-income countries (LMICs) (2,6,10). Moreover, data from cities with extremely high ambient $PM_{2.5}$ levels are limited, restricting our understanding of how chronic exposure affects lung function in the most heavily polluted urban environments [3,6–9]. This lack of data is critical, **as it hinders the development of targeted preventive interventions tailored to the older adults, whose physiological response to extreme pollution may differ significantly from documented patterns in younger cohorts** [10].

Jakarta has repeatedly ranked among the most polluted major cities globally, with annual $PM_{2.5}$ concentrations far exceeding WHO guidelines [2,15]. However, substantial variation in $PM_{2.5}$ levels exists across districts, driven primarily by high-volume traffic emissions and localized urban construction projects [15,16]. This provides a unique opportunity to employ a **natural experiment framework** [17–20]. **Utilizing such high-contrast environmental gradients allows for a more granular understanding of how localized exposure impacts specific demographics. Such evidence is foundational for designing targeted health promotion programs and provides stronger evidence than conventional cross-sectional designs, thereby supporting evidence-informed policy-making.** Unlike standard cross-sectional studies that often lack the temporal depth to establish causal links, this study utilizes the extreme pollution contrast between two distinct regions as a **quasi-gold standard** for environmental observation [17–20]. By enforcing a strict **10-year minimum residency requirement**, we introduce a **quasi-longitudinal perspective** [21,22] that minimizes common pitfalls such as exposure misclassification

and reverse causality, effectively **bridging the ethical and methodological gap** between observational surveys and randomized trials.

To our knowledge, **few studies have evaluated the chronic impact of extreme ambient PM$_{2.5}$ on highly selected healthy older adults using a 10-year stable residency filter within a natural experiment framework.** Furthermore, it applies a quasi-longitudinal approach, providing unique evidence from one of the world's most polluted megacities to **inform evidence-based preventive measures**. To address this evidence gap, we conducted a **natural experiment framework** to compare lung function, expressed as Global Lung Initiative (GLI) z-scores [23,24], between older adult residents of two areas with distinctly different annual PM$_{2.5}$ levels. Our methodology leverages this **environmental contrast** to ensure that the measured lung function **reflects long-term, cumulative impacts** rather than short-term environmental variation. Furthermore, the use of standardized spirometric z-scores (GLI-2012) specifically adjusted for Southeast Asian ethnicity allows for a detailed assessment of the aging lung in this region. **This methodological precision is essential for ensuring that clinical assessments are ethnically and age-appropriately accurate, thereby avoiding the misdiagnosis common in generalized respiratory models and improving the efficacy of primary prevention. By integrating high-resolution environmental data with precise clinical metrics, this study exemplifies the PPH mandate to transform big-data environmental monitoring into actionable, individualized clinical protection.**

The primary research questions were: (1) whether long-term exposure drives lower FEV$_1$ and FVC z-scores among older adults; and (2) whether exposure leads to significant differences in FEV$_1$/FVC z-scores. Drawing from toxicological evidence that chronic PM$_{2.5}$ exposure induces deep alveolar inflammation and systemic oxidative stress, which typically manifests as parenchymal stiffening rather than primary airway obstruction in asymptomatic cohorts, we hypothesized that older adult individuals living in the higher-exposure area would exhibit lower lung volumes (FEV$_1$_z and FVC_z) without overt airflow obstruction, reflecting a subclinical restrictive-like decline, with less pronounced differences in the FEV$_1$/FVC ratio. These findings are expected to provide the granular evidence necessary for **developing robust promotive and preventive public health strategies for aging urban populations.**

## Methods

### Study design

We employed **a natural experiment using a quasi-longitudinal approach** to compare lung function among older adults residing in two areas with markedly different long-term ambient exposure levels. **This design—often regarded as a quasi-gold standard for evaluating environmental exposures—was specifically chosen to bridge the ethical gap where a Randomized Controlled Trial (RCT) is impossible** [25–27]. The substantial between-area variation in levels serves as **an exogenous proxy for random assignment,** allowing for **a population-level comparison that minimizes selection bias. By utilizing these extreme environmental gradients, we created a 'quasi-experimental' condition that enhanced causal inference within the constraints of an observational design while maintaining real-world translational relevance. This design facilitates a granular assessment of environmental impact by moving beyond population-wide averages to evaluate how specific, localized environmental stressors impact a highly vulnerable sub-population.**

**To strengthen the quasi-longitudinal perspective** of this design, we enforced **a strict 10-year minimum residency requirement** for all participants. **This residency threshold ensures that the measured lung function reflects the cumulative, long-term impact of chronic exposure rather than acute environmental shifts.** By establishing **the necessary temporal precedence,** this **approach also significantly reduces the risk of reverse causality** often found in standard cross-sectional studies.

**To our knowledge, this is the first study to utilize such a decadal exposure filter within a natural experiment to assess strictly selected older adults cohort in a high-pollution megacity.**

The study was conducted in accordance with the Strengthening the Reporting of Observational Studies in Epidemiology (STROBE) guidelines (Supplementary Materials), [28,29] to ensure transparent reporting. The completed STROBE checklist is available in the Figshare repository at https://doi.org/10.6084/m9.figshare.31659970. By comparing two geographically distinct populations that are socio-demographically similar but environmentally contrasted, we effectively utilized this **"natural contrast"** to assess the chronic impact of particulate matter on the aging lung while **minimizing confounding** through **strict eligibility criteria** and **frequency matching.**

### Study setting and site selection

The study utilized a two-tiered exposure assessment strategy—a two-stage hybrid approach—to characterize exposure and select research sites by combining macro-level secondary data with micro-level direct environmental sampling. This methodology was employed to ensure a high-contrast exposure gradient while validating area-specific concentrations through ground-truthing [30]. In the first stage (Macro-level Mapping), preliminary mapping was conducted using secondary data from government air quality monitoring stations across the Jakarta and Bandung metropolitan areas to identify regions with the highest and lowest historical levels.

This was followed by a second stage of primary validation (Micro-level Direct Measurement) to capture real-time, area-specific concentrations and minimize the risk of ecological fallacy. Three specific districts within the candidate regions of Jakarta and Bandung were selected for direct on-site monitoring to validate actual ambient conditions within residential neighborhoods. This granular validation is a cornerstone of targeted environmental health surveillance, ensuring that the exposure contrast is not merely assumed from distant monitoring stations but verified at the residential level, thereby reducing exposure misclassification. Based on the synthesis of these multi-level data points, Kedoya (Jakarta) was confirmed as the high-exposure location and Pangalengan (Bandung Regency) as the lower-exposure location. Participants were subsequently assigned to exposure categories using an area-level proxy that reflected the validated chronic ambient conditions of their primary residence.

### Sample size determination

An a priori minimum sample size of 34 participants per group was estimated using a **two-sample comparison of means**, assuming a significance level of α = 0.01 and 95% power [27–29,31]. This calculation employed the standard formula for comparing two independent means:

$$n = \frac{(Z_{\alpha/2} + Z_{\beta})^2 . 2\sigma^2}{(\mu_1 - \mu_2)^2}$$

**Where:**

- n: Minimum sample size required per group.

- $Z_{\alpha/2}$: The Z-score corresponding to the two-tailed significance level (α = 0.01, $Z_{\alpha/2} \approx 2.58$).

- $Z_{\beta}$: The Z-score corresponding to the desired power (1 − β = 0.95, $Z_{\beta} \approx 1.64$).

- μ1 − μ2: The expected minimum mean difference between the high- and low-exposure groups.

- σ: The population standard deviation of the outcome variable ($FEV_1\_z$).

The power calculation was based on an expected mean difference in $FEV_1\_z$-scores (μ1 − μ2) of 0.8 units (assuming a standard deviation (σ) of 1.0), derived from previous similar epidemiological studies on air pollution effects in older adults. To account for potential exclusions, invalid spirometry, and data loss, the recruitment target was increased to at least 60 participants per group [27,31].

## Participants recruitment and eligibility

Data collection was conducted on **May 31, 2025** and **June 14, 2025** across two districts representing high (Kedoya) and low (Pangalengan) exposure gradients. To enhance internal validity and minimize variability in major confounding factors, participants were selected through a multi-layered screening process designed to ensure that observed differences in lung function were primarily attributable to ambient exposure rather than individual health histories or lifestyle choices. This 'ultrapure' cohort selection serves a translational purpose; by isolating the impact of exposure from other major confounders, the findings provide clear, actionable evidence for clinical guidelines and public health protection for the older adults.

Eligibility was determined based on strict inclusion and exclusion criteria. Participants were **included** if they were aged between 65 and 80 years and had maintained permanent residence at their current address for at least ten continuous years. This decadal residency criterion is crucial to ensure that the measured lung function reflects the long-term, cumulative impact of chronic exposure rather than acute environmental changes, effectively establishing a retrospective temporal order between exposure and outcome. Additionally, candidates were required to demonstrate the willingness and physical ability to undergo standardized spirometry and structured interviews, and must fall within the normal Body Mass Index (BMI) range (18.5–22.9 kg/m$^2$) as defined by the WHO classification for Asian populations [32,33]. This BMI restriction was applied to exclude the potential restrictive effects of obesity on lung expansion [34,35].

To further minimize confounding bias, several **exclusion criteria** were rigorously applied. Individuals were excluded if they were current or former active smokers or had significant daily exposure to passive smoking. Potential participants with a known history of chronic respiratory diseases—including COPD, pulmonary fibrosis, uncontrolled asthma, or history of pulmonary tuberculosis—were also excluded, as were those with recent acute respiratory tract infections within the preceding four weeks. Furthermore, the study excluded individuals with severe cardiovascular or systemic conditions, lung cancer, or neuromuscular and cognitive impairments that could interfere with spirometry validity or the ability to follow instructions. Finally, to isolate ambient as the primary factor, individuals with significant occupational exposure to industrial dust, biomass smoke, or chemical irritants, as well as those with communication barriers such as illiteracy, were not included in the final cohort.

## Sampling and final sample

The recruitment and selection process followed a systematic two-stage sampling approach to ensure a high-quality, comparable study population (See Fig 1 – STROBE Flowchart). In the first stage (area level), purposive selection was used to identify regions and specific sites (Kedoya and Pangalengan) with extreme contrast through macro-monitoring and ground-truthing. In the second stage (individual level), a sampling frame of 245 potential candidates was established through registries at community health posts (Posbindu). These candidates underwent preliminary pre-screening to verify basic eligibility based on age (65–80 years), never-smoking status, and a 10-year stable residency requirement.

From this registry, a stratified random sampling procedure was employed to invite 60 candidates from each site (total n = 120) for formal clinical assessment to minimize selection bias. A total of 123 individuals attended the clinical phase (68 from Kedoya and 55 from Pangalengan). These candidates underwent a rigorous one-day comprehensive clinical screening, comprising structured interviews, physical examinations, and Electrocardiography (ECG) to rule out cardiovascular abnormalities. To enhance comparability and neutralize potential confounders—such as age, gender, education level, and occupation—a frequency matching procedure was applied. After strictly adhering to the ATS/ERS 2019 quality standards for spirometry [36], the final analytical sample consisted of 101 participants (54 from the higher-exposure area and 47 from the lower-exposure area).

## Research variables and operational definitions

This study involved several key variables classified into independent, dependent, and covariate variables. The independent variable was the level of chronic fine particulate matter (PM$_{2.5}$) exposure, represented by the participants' residential

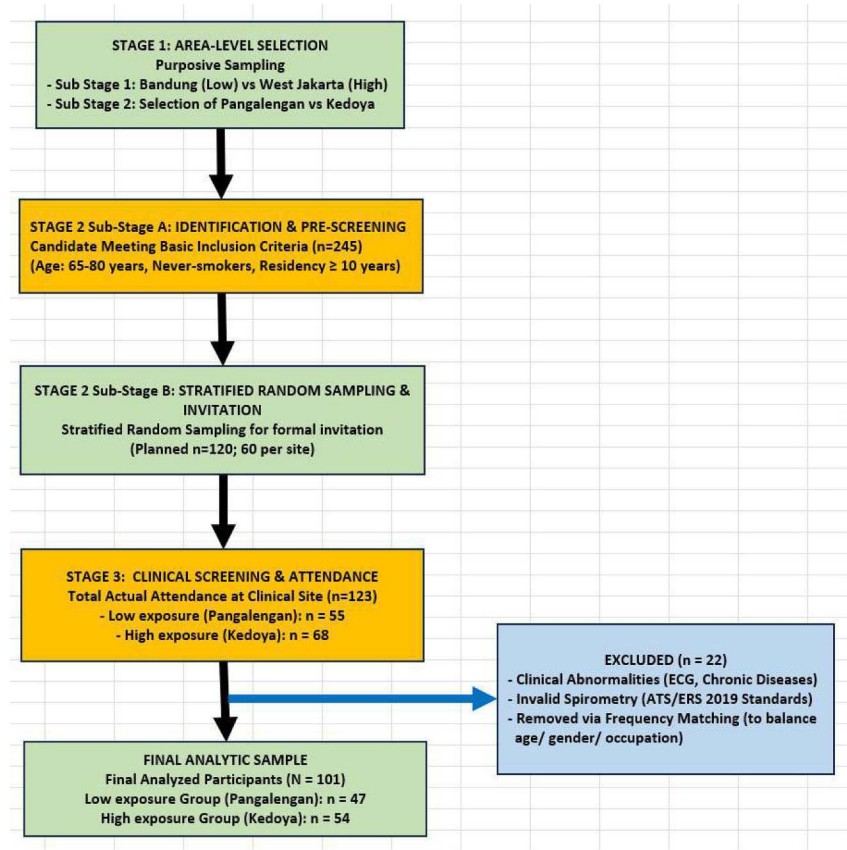

**Fig 1. STROBE Flowchart of the two-stage sampling and participant selection process.** The diagram illustrates the recruitment stages: Stage 1 (Area-level) involved purposive site selection (Sub-Stages 1–2), and Stage 2 (individual-level) involved participant identification, pre-screening, stratified random sampling for formal invitation. The final analytical sample was established after clinical validation, strict adherence to ATS/ERS 2019 standards, and frequency matching to ensure healthy, comparable groups.

location, with the Kedoya area categorized as high exposure and Pangalengan as low exposure. A primary focus of this study was the assessment of lung function as a continuous physiological measure rather than a binary clinical diagnosis. This approach increases statistical power and allows for the detection of subclinical declines in respiratory health, facilitating proactive primary prevention by identifying 'at-risk' individuals before overt clinical disease manifests.

The primary outcomes were lung function indices expressed as FEV1 and FVC z-scores, calculated using the Global Lung Initiative (GLI) 2012 Southeast Asian reference equations to account for age, sex, and height. Analyzing these as continuous variables enables the detection of subtle, dose-dependent shifts in lung function associated with chronic exposure. Secondary outcomes included the $FEV_1$/FVC z-score and the prevalence of lung function impairment.

Participants were classified based on the Lower Limit of Normal (LLN), defined as a z-score threshold of −1.645. Clinical airflow obstruction was identified by an FEV1/FVC z-score below this threshold. Conversely, participants with an $FEV_1$/FVC z-score ≥ −1.645 accompanied by reduced $FEV_1$ and/or FVC z-scores below −1.645 were classified as having a non-obstructive spirometric impairment, characterized by reduced lung volumes with a preserved ratio. This z-score–based approach provides a more precise classification for our older adult cohort than fixed percentage thresholds by effectively accounting for age-related physiological changes and enhancing the detection of early, subclinical lung function impairment [36].

Covariate variables were collected for descriptive analysis, frequency matching, and statistical adjustment. These included age (years), gender (male or female), and height (cm). Body Mass Index (BMI) was calculated as weight in kilograms divided by the square of height in meters ($kg/m^2$). In accordance with the inclusion criteria, all participants were within the WHO-defined normal range for Asian populations. BMI served both for frequency matching and as a covariate in multivariable models, with an independent sample t-test confirming no statistically significant difference in mean BMI between the two exposure groups to ensure baseline comparability of nutritional status. Socioeconomic status (SES) was captured through education level—categorized into low (no schooling/primary), middle (junior/senior high), and higher education (diploma/degree)—and occupation, which was grouped ordinally into three levels: (1) unemployed, laborer, farmer, or small trader; (2) employee; and (3) professional.

## Research instruments

Data collection was facilitated through a suite of validated instruments designed to capture demographic, socioeconomic, and physiological variables. A pre-tested structured questionnaire was utilized to gather respondent identity, socioeconomic status (including education and occupation), and a detailed 10-year residency history. To ensure the 'ultrapure' nature of the cohort, the questionnaire also screened for histories of chronic or acute lung diseases and environmental confounding factors, such as passive smoking and biomass exposure. All measurements were recorded on a standardized observation sheet, which included height measurements taken with a calibrated stadiometer to the nearest centimeter. Ethical adherence was documented through signed informed consent forms, which were obtained after participants received comprehensive information regarding the study's objectives and procedures.

Pulmonary function was assessed using a portable digital spirometer to measure Forced Expiratory Volume in 1 second ($FEV_1$), Forced Vital Capacity (FVC), and the $FEV_1/FVC$ ratio. The device maintained a volume accuracy of 3% and a flow range of 0–16 L/s, with calibration performed before each session to ensure data integrity. All measurements were conducted by trained personnel in strict accordance with the American Thoracic Society/European Respiratory Society (ATS/ERS) standards. Participants received standardized instructions and demonstrations prior to the examination, with maneuvers repeated until at least two consistent, high-quality results were obtained. To further validate site conditions and technical reporting, visual documentation was utilized for location-specific field notes and activity verification, while strictly maintaining participant anonymity.

## Data collection methods

Data collection was conducted directly at the two research locations by a trained survey team comprising enumerators, medical personnel, and field coordinators. The fieldwork utilized standardized procedures and instruments, beginning with a community-based recruitment process. After passing initial pre-screening and prior to any clinical procedures, participants received a comprehensive explanation of the study and provided written informed consent. Face-to-face interviews were then conducted using a piloted structured questionnaire to collect demographic data, socioeconomic factors, residency history, and detailed medical histories related to the exclusion criteria. To minimize information and observer bias, all interviews were conducted by trained enumerators who were blinded to the participants' exposure status.

Participants subsequently underwent standardized physical and supporting examinations. Anthropometric measurements, including body weight and height (measured using a portable stadiometer), were taken to confirm that all participants met the normal BMI inclusion criteria (18.5–22.9 $kg/m^2$). This was followed by a cardiorespiratory assessment—encompassing blood pressure, heart rate, respiratory rate, oxygen saturation, and lung/heart auscultation—and an Electrocardiogram (ECG) to exclude severe cardiovascular conditions. All clinical examinations were performed by medical personnel who were blinded to the geographic exposure classification of the participants.

Pulmonary function was measured using a portable digital spirometer in strict accordance with the American Thoracic Society (ATS) and European Respiratory Society (ERS) standards [35]. Participants performed maneuvers in a seated

position, and measurements were repeated until at least dua consistent best results were obtained, with a difference in and between maneuvers not exceeding 150 mL. To ensure data validity across the geographically distinct study sites of Kedoya and Pangalengan, the following protocols were implemented. First, to account for significant differences in temperature and atmospheric pressure between the coastal Jakarta area and the highland Bandung region, the spirometer was equipped with an internal BTPS (Body Temperature, Ambient Pressure, Saturated with water vapor) sensor. This sensor automatically calibrated and adjusted measured lung volumes based on ambient conditions at each site, ensuring that the results reflected true physiological capacity. This environmental compensation ensured the precision of the biological signal across disparate geographic altitudes. Second, raw lung function values were converted into z-scores using the Global Lung Initiative (GLI) 2012 Southeast Asian-adjusted reference equations to account for ethnic-specific lung morphology. This approach minimized bias related to age, height, and ethnicity, while all spirometry technicians remained blinded to participants' exposure status to ensure objective outcome measurement.

### Air pollution exposure data

Air pollution exposure data ($PM_{2.5}$ levels) were obtained from long-term environmental monitoring data at the participants' residential areas, which served as the basis for categorizing participants into the high and low exposure groups.

### Data recording and quality control

All data were recorded on standardized worksheets. Data quality control involved review by a field supervisor, reconfirmation of incomplete or questionable data, and double data entry to minimize input errors and ensure accuracy.

### Outcome measures

The primary outcomes of this study were standardized lung function indices, which facilitate an accurate comparison between individuals across different ages, heights, and genders. We focused on three key parameters to assess pulmonary health: Forced Expiratory Volume in 1 second ($FEV_1$), Forced Vital Capacity (FVC), and the $FEV_1$/FVC ratio, the latter of which is used to identify obstructive ventilatory defects. To minimize the bias inherent in using absolute values or percentage of predicted values—which can vary significantly in older adultss—all raw measurements were **converted into z-scores** using the Global Lung Function Initiative (GLI-2012) reference equations.

   A z-score indicates how many standard deviations an individual's measurement deviates from the mean predicted value of a healthy population. This standardization ensures that differences in lung function between the high and low exposure groups are not confounded by individual anthropometric characteristics, including age, height, gender, and ethnicity. For the clinical categorization of impairment, participants were classified as having a ventilatory defect if their $FEV_1$_z, FVC_z, or $FEV_1$/FVC_z-score fell below the Lower Limit of Normal (LLN). Following international diagnostic standards, the LLN was defined as a **z-score of less than −1.645**, which corresponds to the 5th percentile of the healthy reference population.

### Data quality and handling of missing data

To ensure high data integrity, a complete case analysis was adopted. Given the direct supervision during data collection, missingness was minimal. Any participants with invalid spirometry or incomplete responses were replaced in real-time by new recruits from the same strata to maintain the required statistical power.

### Statistical analysis

Participant characteristics were summarized using descriptive statistics. Initially, the effectiveness of the frequency matching procedure was verified by comparing baseline characteristics between the high and lower-exposure groups using

Independent Samples t-tests (or Mann-Whitney U tests) for continuous variables and Chi-square tests for categorical variables.

Data normality was assessed using the **Shapiro-Wilk test**, as it is more sensitive and robust for the current sample size per group ($n < 100$), supplemented by visual inspection of Q-Q plots. Homogeneity of variances was evaluated using **Levene's test**. Group differences between exposure areas were assessed using **Independent Samples t-tests** for variables meeting parametric assumptions or nonparametric alternatives (**Mann-Whitney U**) for variables that violated normality or homogeneity assumptions.

**Multivariable linear regression** models were fitted to estimate the association between $PM_{2.5}$ exposure category and each lung function z-score ($FEV_1\_z$, $FVC\_z$, and $FEV_1/FVC\_z$-score). Models were adjusted for age, sex, body mass index, socioeconomic indicators. Although lung function outcomes were expressed as GLI-2012 z-scores, **age, sex, and height were additionally included as covariates** to account for **potential residual confounding** and to **ensure conservative model specification.** Results are reported as adjusted mean differences (Unstandardized B coefficients) with 95% confidence intervals. Model assumptions were strictly verified using the **Kolmogorov–Smirnov test to confirm the normality of unstandardized residuals (with a significance threshold of $p > 0.01$), and supplemented by visual inspection of residual Q-Q plots. Multicollinearity** was assessed using the **Variance Inflation Factor (VIF < 5.0). Model stability** and influential outliers were screened using **Cook's distance (threshold < 0.5)**, and **Mahalanobis distance with the latter evaluated against the χ2 critical value for the corresponding degrees of freedom ($p < 0.001$).** All model assumptions, including normality of residuals and homoscedasticity, were strictly verified to ensure the validity of parametric estimates.

Statistical analyses were conducted using SPSS version 26. All hypothesis tests were evaluated using a two-tailed significance level of **α = 0.01** and **power = 95%, providing the high-precision evidence necessary for translational health applications. While the minimum required sample size to detect large effects was 68 ($n = 34$ per group), our final cohort of $n = 101$ provided superior statistical power (> 95%), ensuring high-precision evidence for evaluating decadal exposure impacts.**

## Ethical considerations

The study was conducted in accordance with the **Declaration of Helsinki** and **approved by the Ethical Review Committee of the Faculty of Medicine, Universitas Trisakti** under ethical permission number 007/KER/FK/04/2025. Prior to participation, all individuals provided **written informed consent**. To ensure genuine autonomy among the older adults participants, the consent process included a comprehensive verbal explanation of the study's objectives and the non-invasive nature of the procedures.

Researchers ensured that all technical terms were explained in **lay language**, and for those with age-related visual difficulties, the form was read aloud to guarantee full comprehension before signing. Participation was strictly voluntary, with the right to withdraw at any time without consequence. Each consent form was signed by the participant and **witnessed by an impartial third party** present during the process. Individuals who were illiterate or could not speak Indonesian were excluded to ensure the integrity of the informed consent process. As the study focused exclusively on the older adults, no minors were involved.

To safeguard privacy, all data were **double-anonymized** using unique alphanumeric codes; no identifying information was stored in the final analytic dataset. All clinical examinations were conducted in private settings to maintain the dignity and comfort of the participants.

## Results

### A. Participant flow and selection

The flow of participants throughout the study, detailing the recruitment process, application of inclusion/exclusion criteria, and the final analytic sample size, is illustrated in Fig 1. A total of 123 older adults residents were initially screened. After

applying eligibility criteria (e.g., age 65–80 years, minimum 10-years residency, no history of smoking) and excluding 22 participants, the final analytic sample consisted of 101 older adults. **The exclusions were due to clinical abnormalities identified during screening (n = 14), invalid spirometry maneuvers that did not meet the ATS/ERS 2019 quality standards (n = 3), and the frequency matching procedure to ensure group comparability (n = 5). The complete selection flow is illustrated in Fig 1.** Specifically, **47 participants** from the lowest-exposure Pangalengan group and **54 participants** from the highest-exposure Kedoya group were included in the final analysis. The Pangalengan group consisted of **37 older adults women and 10 older adults men**, while the Kedoya group comprised **47 older adults women and 7 older adults men.**

## B. Study setting and exposure assessment

To establish the environmental contrast required for this study, $PM_{2.5}$ concentration measurements were conducted through a two-stage hybrid exposure assessment to establish the environmental contrast between the two study areas. In the first stage, we reviewed longitudinal ambient $PM_{2.5}$ data from government-operated monitoring stations, which provided daily records to confirm the long-term historical pollution profiles of the two regions. These records consistently indicated substantially higher ambient $PM_{2.5}$ levels in Jakarta compared to the Bandung area (characterized by generally low $PM_{2.5}$ levels).

In the second stage, to validate neighborhood-level conditions during the study period, primary measurements were conducted at six preliminary locations (three in Jakarta and three in Bandung). These validation measurements were performed using calibrated sensors over 24-hour sampling periods by an independent certified air quality monitoring company. Based on these data, Kedoya in Jakarta area was selected as the highest-exposure area and Pangalengan in Bandung area as the lowest-exposure area. To ensure the robustness of the area-level exposure proxy, all participants were recruited from a single administrative sub-district (*kelurahan*) immediately surrounding these validated sites. The measurement results for AQI (Air Quality Index), $PM_{2.5}$, and $PM_{10}$ from all six preliminary locations are presented in Table 1.

## C. Participant characteristics

Baseline demographic and clinical characteristics are detailed in Table 2. The two exposure groups were generally comparable across most variables, confirming the **effectiveness of the frequency matching procedure** outlined in the Methods section. Mean age in the low-exposure group (Group 1) was 71.32 years vs 71.52 years in the high exposure group (Group 2). The difference in mean age (Mean Differences=−0.199 years) was not statistically significant (t=−0.317, p = 0.747). Homogeneity of variance for age was also confirmed by Levene's test (F = 5.075, p = 0.026, consistent with α = 0.01 as the criterion).

Similarly, the distribution of gender was comparable, with the majority being female in both groups (Pangalengan: 78.72% female vs. Kedoya: 87.04% female, χ2 test p = 0.265). The lack of significance difference in mean age and gender

Table 1. AQI (Air Quality Index), $PM_{2.5}$ and $PM_{10}$ Concentrations in Preliminary Measurement Areas.

| Area | Location | AQI | PM2.5 (µg/m³) | PM10 (µg/m³) |
|---|---|---|---|---|
| Bandung | Pangalengan | 43 | 10 | 17 |
| | Leuwi Panjang | 179 | 36 | 87 |
| | Padalarang | 191 | 40 | 94 |
| Jakarta | Kedoya | 219 | 57 | 156 |
| | Cilandak Barat | 216 | 54 | 151 |
| | Mangga Dua | 209 | 45 | 127 |

**Table 2. Demographic and Anthropometric Characteristics of Study Participants by Data Collection Location.**

| Characteristics | Pangalengan | Kedoya | Total |
|---|---|---|---|
| Gender: | | | |
| - Male | 10 (21.28%) | 7 (12.96%) | 17 (16.83%) |
| - Female | 37 (78.72%) | 47 (87.04%) | 84 (83.17%) |
| Number of Participants | 47 (100%) | 54 (100%) | 101 (100%) |
| Education: | | | |
| - Low: Not attending school, Elementary school | 37 (72.34%) | 38 (70.37%) | 75 (74.26%) |
| - Middle: Junior and Senior High School | 9 (19.14%) | 14 (25.93%) | 23 (22.77%) |
| - High: diploma, bachelor | 1 (2.13%) | 2 (3.70%) | 3 (2.97%) |
| Occupation: | | | |
| - unemployed, laborers, farmers, small traders | 47 (100%) | 54 (100%) | 101 (100%) |
| - employee | 0 (0%) | 0 (0%) | 0 (0%) |
| - professional | 0 (0%) | 0 (0%) | 0 (0%) |
| Age | Mean = 71.32, SD = 2.59 | Mean = 71.52, SD = 3.57 | |
| Height | Mean = 147.53, SD = 7.21 | Mean = 151.30, SD = 6.04 | |
| Normal Body Mass Index (18.5–22.9 kg/m2) | Mean = 20,74, SD = 0.80 | Mean = 20,80, SD = 0.70 | |
| Ethnicity | Southeast Asia | Southeast Asia | |

distribution demonstrated that the groups were well-balanced for these important covariates. No data were missing for lung function variables or covariates.

The anthropometric profile, specifically Body Mass Index (BMI), also showed no significant difference between the two groups. The mean BMI in the low-exposure group was **20.74 kg/m²** compared to **20.80 kg/m²** in the high-exposure group (**Mean Difference = −0.066 kg/m²**). This difference was not statistically significant (**t = −0.441, p = 0.661**). Homogeneity of variance for BMI was confirmed by Levene's test (**F = 1.197, p = 0.277**), further demonstrating that the groups were well-balanced regarding nutritional status.

Clinical and physiological baseline assessments showed no significant differences between the low-exposure and high-exposure groups. The median respiratory rate was 19.0 breaths/min for both groups (p = 0.032), while oxygen saturation remained stable with a median of 96.0% (p = 0.947). Other vital signs, including blood pressure, heart rate, and body temperature, were also within normal clinical ranges and did not differ significantly between groups. Detailed individual-level data for all clinical parameters are provided in S1 Table.

To compare the characteristics of the study subjects between the Pangalengan group (Group 1) and the Kedoya group (Group 2), a series of statistical tests were conducted, including normality of distribution tests, homogeneity of variances tests, and mean or median comparison tests. A comprehensive summary of these test results is presented in Table 3.

## D. Lung function differences between exposure groups

Participants residing in the higher exposure area exhibited lower mean $FEV_1\_z$-scores and FVC_z-scores compared with those living in the lower-exposure area. In contrast, the $FEV_1$/FVC z-score remained comparable between the two groups.

1. Comparison **of** Z-Scores (Adjusted Parameters).

For the z-scores ($FEV_1\_z$ and FVC_z), which adjust for age, gender, and height, the analysis revealed the following: **$FEV_1\_z$ (Non-parametric Test):** Due to the violation of the normality assumption in group 1 (p = 0.001) and unequal variances (Levene's p < 0.001), the Mann-Whitney U test was performed. The test revealed a **highly statistically significant difference** (p < 0.001). The lower-exposure group showed a higher distribution (Mean Rank = 64.63; Mean = −0.45) compared to the higher-exposure group (Mean Rank = 39.14; Mean = −1.89), with a Mean Rank difference of +25.49. Notably,

**Table 3. Summary of Statistical Test Results for Inter-Group Characteristic Comparisons.**

| Variable Compared | Normality Test Group 1/Group 2) | Variance Homogeneity (Levene's Sig.) | Comparison Test Used | Test p-value | Conclusion |
|---|---|---|---|---|---|
| Test of Difference in Proportions of Groups 1&2 | N/A (Categorical) | N/A | Chi-Square Goodness of Fit Test | 0.486 | No significance different |
| Test of Difference in Gender Proportion* Group 1&2 | N/A (Categorical) | N/A | Chi-Square of Independence Test | 0.265 | No significance different |
| Test of Difference in Education Proportion* Group 1&2 | N/A (Categorical) | N/A | Chi-Square (Monte-Carlo exact) | 0.663 | No significance different |
| Age: | | | | | |
| Group 1 | 0.146 (Normal Distribution) | 0.026 (Homogeneous) | Welch/Brown-Forsythe | 0.747 | No significance different |
| Group 2 | 0.102 (Normal Distribution) | | | | |
| Height: | | | | | |
| Group 1 | 0.175 (Normal Distribution) | 0.337 (Homogeneous) | Welch/Brown-Forsythe | 0.006 (Welch/Brown-Forsythe) | Significance different |
| Group 2 | 0.162 (Normal Distribution) | | | | |
| Body Mass Index: | | | | | |
| Group 1 | 0.536 (Normal Distribution) | 0.277 (Homogenous) | Independent Sample t-test | 0.661 | No significance different |
| Group 2 | 0.111 (Normal Distribution) | | | | |

**Note:** Group 1: Pangalengan; Group 2: Kedoya; N/A: Not Applicable.

while the lower-exposure group remained within the normal range, the mean of the higher-exposure group fell below the clinical threshold of **−1.645**, indicating a systematic shift toward impaired lung function.

**FVC_z (Parametric Test):** As the α = 0.01 significance level, FVC_z satisfied the assumption for parametric analysis. Both groups followed a normal distribution (Shapiro-Wilk p = 0.054 and p = 0.989) and demonstrated homogeneity of variances (Levene's p = 0.051). Consequently, an **Independent Samples t-test** was performed. The test indicated a **highly statistically significant difference** between the two exposure groups (p < 0.001).

- The **Mean Difference (μ1−μ2)** was calculated as +1.2111 **z-score units** (Mean 1 = −1.2409; Mean 2 = −2.4520), corresponding to a **large effect size** (**Cohen's d=1.21**). This indicated that the average **FVC_z score** in the lower-exposure group was **1.21 Standard Deviations (SDs) higher** than the higher-exposure group.

**FEV₁/FVC_z Ratio:** Consistent with the primary findings, the Mann-Whitney U test for the $FEV_1$/FVC_z ratio showed **no statistically significant difference** between the two groups (p = 0.464), with a similar distribution of ranks observed in the lower-exposure group (**Mean Rank = 53.29**) compared to the higher-exposure group (**Mean Rank = 49.01**). Both groups maintained average values (+1.96 and +1.57, respectively) well above the clinical threshold for obstruction (Lower Limit of Normal = −1.645). This result is consistent with the study's recruitment process, which excluded individuals with known respiratory diseases or overt symptoms of breathlessness, thereby ensuring a study population of 'apparently healthy' older adults. Consequently, the proportional relationship between and remained preserved, confirming an absence of airway obstruction even in the higher-exposure group. **Notably, both the lower-exposure group (+1.96) and the higher-exposure group (+1.57) maintained mean values well above the clinical threshold for obstruction (LLN=−1.64). This result aligns with the study's recruitment process, which excluded individuals with known respiratory diseases or overt symptoms of breathlessness to ensure a population of 'apparently healthy' older**

**adults. Consequently, while absolute volumes (FEV1_z and FVCz) were significantly reduced, the proportional relationship between them remained preserved, confirming an absence of airway obstruction even in the higher-exposure group.**

2. Consistency in Actual Parameters

This pattern was consistently mirrored in the analysis of actual lung function parameters ($FEV_1$_actual and FVC_actual):

- The mean $FEV_1$_actual was significantly lower in the high-exposure group (Independent -test, $p < 0.001$).
- The median FVC_actual also showed a statistically significant difference (Mann-Whitney U test, $p = 0.002$).
- Correspondingly, no significant difference was detected in the median $FEV_1$/FVC_actual ($p = 0.687$).

These findings strongly suggest that differences in $PM_{2.5}$ exposure were primarily associated with reductions in absolute **lung volumes** ($FEV_1$_z and FVC_z), rather than an alteration in the underlying **airway resistance pattern** (as indicated by the non-significant $FEV_1$/FVC_z ratio). Overall, chronic exposure to higher levels was associated with a pattern consistent with reduced lung volumes without airflow obstruction, as indicated by significantly lower and scores but preserved ratios.

The complete comparison results for both actual and z-scores lung function parameters are detailed in Table 4.

## E. Correlation analysis between covariates and lung function

To explore potential confounders and inform the subsequent multivariable model, bivariate correlations were examined between selected participant characteristics (covariates) and lung function z-scores. The complete correlation matrix, including the results of both Pearson (r) and Spearman (ρ) tests, is summarized in Table 5.

The key finding confirmed the inverse association observed in the descriptive analysis: Ambient $PM_{2.5}$ exposure demonstrated a significant and moderate negative correlation with both $FEV_1$_z (Pearson $r = -0.469$, $p < 0.001$) and FVC_z (Pearson $r = -0.462$, $p < 0.001$). This suggests that higher levels are associated with lower lung volumes relative to reference

**Table 4. Summary of Statistical Tests for $FEV_1$, FVC, and $FEV_1$/FVC Variables.**

| Variable | Group | Normality Test (Shapiro-Wilk Sig.) | Homogeneity of Variance (Levene's Sig.) | Tests Performed | Sig. Test Results | Conclusion (Differences Between Groups) |
|---|---|---|---|---|---|---|
| $FEV_1$_actual | 1 | 0.019 (Normal) | 0.672 (Homogeneous) | Independent Samples t-test | <0.001 | Significant Difference |
| | 2 | 0.058 (Normal) | | | | |
| FVC_actual | 1 | 0.008 (Non Normal) | 0.634 (Homogeneous) | Mann-Whitney U | 0.002 | Significant Difference |
| | 2 | 0.520 (Normal) | | | | |
| $FEV_1$FVC_actual | 1 | <0.001 (Non Normal) | 0.058 (Homogeneous) | Mann-Whitney U | 0.687 | No Significant Difference |
| | 2 | <0.001 (Non Normal) | | | | |
| $FEV_1$_z | 1 | 0.001 (Non Normal) | 0.000 (Not Homogeneous) | Mann-Whitney U | <0.001 | Significant Difference |
| | 2 | 0.369 (Normal) | | | | |
| FVC_z | 1 | 0.054 (Normal) | 0.051 (Homogeneous) | Independent Samples t-test | <0.001 | Significant Difference |
| | 2 | 0.989 (Normal) | | | | |
| $FEV_1$FVC_z | 1 | <0.001 (Non Normal) | 0.017 (Homogeneous) | Mann-Whitney U | 0.464 | No Significant Difference |
| | 2 | <0.001 (Non Normal) | | | | |

**Abbreviations:** $FEV_1$ – Forced Expiratory Volume in 1 second; FVC – Forced Vital Capacity; $FEV_1$/FVC – Ratio of $FEV_1$ to FVC; $FEV_1$_actual – Measured absolute value of $FEV_1$ (in liters); FVC_actual – Measured absolute value of FVC (in liters);$FEV_1$/FVC_actual – Measured absolute ratio of $FEV_1$ to FVC; $FEV_1$_z – Standardized Z-score of $FEV_1$ (adjusted for age, gender, and height); FVC_z – Standardized Z-score of FVC (adjusted for age, sex, and height); $FEV_1$/FVC_z – Standardized Z-score of the $FEV_1$/FVC ratio; Sig. – Significance (p-value); Levene's Sig. – p-value from Levene's test for homogeneity of variances.

**Table 5. Pearson Correlation and Spearman's rho Matrix between PM$_{2.5}$ Exposure, Covariates and Lung Function Variables in the Older adults.**

| Variables | Pearson r Correlation Coeff. (p) | Spearman's ρ Correlation Coeff. (p) | Conclusion |
|---|---|---|---|
| PM2.5 vs FEV$_1$_z | −0.469 (p < 0.001) | −0.436 (p < 0.001) | Significant and moderate negative correlation |
| PM2.5 vs FVC_z | −0.442 (p < 0.001) | −0.415 (p < 0.001) | Significant and moderate negative correlation |
| PM2.5 vs FEV$_1$/FVC_z | −0.136 (p = 0.176) | −0.073 (p = 0.467) | Not statistically significant |
| Gender vs FEV1_z | 0.078 (p = 0.439) | 0.058 (p = 0.564) | Not statistically significant |
| Gender vs FVC_z | 0.099 (p = 0.324) | 0.118 (p-0.240) | Not statistically significant |
| Gender vs FEV$_1$/FVC_z | 0.039 (p = 0.699) | 0.031 (p = 0.759) | Not statistically significant |
| Age vs FEV$_1$_z | −0,111 (p = 0.269) | −0,136 (p = 0.175) | Not statistically significant |
| Age vs FVC_z | −0,118 (p = 0.239) | −0.122 (p = 0.223) | Not statistically significant |
| Age vs FEV$_1$/FVC_z | −0.085 (p = 0.397) | −0,138 (p = 0.168) | Not statistically significant |
| Height vs FEV$_1$_z | −0,238 (p = 0.016) | −0,124 (p = 0.218) | Not statistically significant |
| Height vs FVC_z | −0,273 (p = 0.006) | −0,184 (p = 0.066) | Significant and weak negative correlation |
| Height vs FEV$_1$/FVC_z | 0.016 (p = 0,876) | 0.065 (p = 0.517) | Not statistically significant |
| Education vs FEV$_1$_z | 0.086 (p = 0.393) | 0.162 (p = 0.106) | Not statistically significant |
| Education vs FVC_z | 0.046 (p = 0.649) | 0.095 (p = 0.345) | Not statistically significant |
| Education vs FEV$_1$/FVC_z | 0.184 (p = 0.066) | 0.146 (p = 0.146) | Not statistically significant |
| BMI vs FEV$_1$_z | 0.214 (p = 0.031) | 0.146 (p = 0.145) | Not statistically significant |
| BMI vs FVC_z | 0.099 (p = 0.327) | 0.078 (p = 0.440) | Not statistically significant |
| BMI vs FEV$_1$/FVC_z | 0.210 (p = 0.035) | 0.150 (p = 0.135) | Not statistically significant |

**Note:** Significance evaluated at $p < 0.01$ (two-tailed); $p \geq 0.01$ was interpreted as not statistically significant (NS). Conclusions were based on the appropriate correlation test according to data distribution: Spearman's rho was applied when at least one variable was non-normally distributed, and Pearson's correlation was applied when both variables were normally distributed. Magnitude guidelines (absolute coefficient): **weak** = 0.10–0.29; **moderate** = 0.30–0.49; **strong** ≥ 0.50. The sign indicates direction (negative/positive). See Table 4 for variable abbreviations.

values. Consistent with the group comparison results, PM$_{2.5}$ exposure was not statistically significantly associated with the FEV$_1$/FVC_z ratio (Pearson r=−0.136, p = 0.176).

Regarding the covariates, Age, Gender, Education, and Body Mass Index (BMI) showed no statistically significant correlations with any of the lung function outcomes when evaluated at the prespecified significance level of p < 0.01. However, Height showed a significant, weak negative correlation with FVC_z (Pearson r=−0.273, p = 0.006), suggesting a potential weak confounding effect that should be controlled for in the regression analysis.

## F. Multivariable linier regression

**Multivariable linear regression** was employed to determine the independent association between exposure (group status) and lung function Z-scores. The models were **adjusted for potential confounders**, including age, gender, education, height, and BMI. The results of the three adjusted models are summarized in Table 6.

## Model for FEV$_1$_z

The overall model for FEV$_1$_z was **statistically significant** (F(6,94)=7.189, p < 0.001), explaining approximately **31.5%** of the variance in FEV$_1$_z ($R^2$ = 0.315). In this adjusted model, **residence in the high-exposure area emerged as the only significant driver** of lower FEV$_1$_z (Unstandardized Coefficient B=−1.434, p < 0.001). This indicates that, after controlling for all listed covariates, participants in the high PM$_{2.5}$ area had an FEV$_1$_z score that was 1.434 units lower than those in the low PM$_{2.5}$ area. While BMI showed a positive association (B = 0.443, p = 0.015), it did not meet the stringent

**Table 6. Summary of Multivariable Linear Regression Analysis.**

| Variables | R² (Adj. R²) | F (df) | p-value Model | Prediktor Signifikan (p<0.01) | VIF Maks | Tolerance Min | Std. Residual Range | Cook's D Max | Mahalano-bis D Max | Remarks |
|---|---|---|---|---|---|---|---|---|---|---|
| FEV$_1$_z | 0.315 (0.271) | F(6, 94) = 7.189 | < 0.001 | Group (PM$_{2.5}$), p<0.001 | 1.562 | 0.64 | −2.058 to +2.676 | 0.149 | 20.338 | Model significant; assumptions met; no outliers or multicollinearity detected. |
| FVC_z | 0.259 (0.212) | F(6, 94) = 5.475 | < 0.001 | Group (PM$_{2.5}$), p<0.001 | 1.562 | 0.64 | −1.869 to 2.282 | 0.123 | 20.338 | Model significant; assumptions met; no outliers or multicollinearity detected. |
| FEV$_1$FVC_z | 0.110 (0.054) | F(6, 94) = 1.943 | 0.082 | None | 1.562 | 0.64 | −2.636 to +1.518 | 0.104 | 20.338 | Model not significant; all predictors p≥0.01; no assumption violations detected. |

**Note:** All variable abbreviations (e.g., FEV$_1$_actual, FEV$_1$_z, FVC_z) are defined in Table 4. Statistical terms (e.g., R², Adj. R², VIF, Cook's D, Mahalanobis D) follow standard statistical nomenclature.

significance threshold of 0.01. The model's validity was confirmed by a maximum Cook's distance of 0.149 and VIF values below 1.6, indicating no influential outliers or multicollinearity issues.

### Model for FVC_z

Similarly, the model for FVC_z was also **statistically significant** (F(6,94)=5.475, p<0.001), accounting for **25.9%** of the variance (R²=0.259). The PM$_{2.5}$ exposure group was the **sole significant predictor** (Unstandardized Coefficient B = −1.145, p<0.001), demonstrating that the impact of higher exposure on reduced lung volume remains robust after adjustment.

### Model for FEV$_1$/FVC_z

In contrast, the model for FEV$_1$/FVC_z was **not statistically significant** (F(6,94)=1.943, p=0.082), accounting for only 11.0% of the variance (R²=0.110). Consistent with the unadjusted results, **no predictor variable**, including PM$_{2.5}$ exposure (B=−0.560, p=0.071), showed a statistically significant association with the FEV$_1$/FVC_z ratio after adjustment.

Overall, these findings indicate that long-term exposure to elevated PM$_{2.5}$ levels is significantly associated with reduced lung volumes (FEV$_1$_z and FVC_z) but not with evidence of obstructive impairment (indicated by the FEV$_1$/FVC_z ratio) in this older adults population.

### Assessment of model assumptions

Prior to final interpretation, key assumptions of the multivariable linear regression models were assessed. Multicollinearity was confirmed to be minimal, with Variance Inflation Factor (VIF) values remaining well below the threshold of 5 for all models. Furthermore, residual normality was formally tested for each model, with results summarized in Table 7.

Based on the strict α=0.01 criterion, the residuals for the FEV$_1$_z model met the criteria for normality (K–S p=0.176; S–W p=0.018). Similarly, the FVC_z model residuals demonstrated strong normality (K–S p=0.200; S–W p=0.114), validating the use of parametric tests in these models. While the residuals for the FEV$_1$/FVC_z ratio model showed a deviation from formal normality (S–W p=0.001), they were considered approximately normal given the large sample size (n=101). As shown in Table 6, the multivariable linear regression models demonstrated high stability. **The sensitivity analysis**, integrated through the assessment of **Cook's Distance (max<0.15) and Mahalanobis Distance (max=20.338), confirmed that no influential outliers biased the model estimates**, further **validating the robustness**

**Table 7. Residual Normality Tests for Multivariable Linear Regression Models.**

| Dependent Variable | n | Kolmogorov–Smirnov (Stat; p) | Shapiro–Wilk (Stat; p) | Normality Verdict* | Brief Interpretive Note |
|---|---|---|---|---|---|
| $FEV_1$_zRES_8 | 101 | (K–S Stat); 0.176 | (S–W Stat); 0.018 | Accepted | Residuals follow a normal distribution at α = 0,01. |
| FVC_zRES_9 | 101 | (K–S Stat); 0.200 | (S–W Stat); 0.114 | Accepted | High degree of normality; no transformation required. |
| $FEV_1$/FVC_z RES_10 | 101 | (K–S Stat); 0.020 | (S–W Stat); 0.001 | Approximately normal | n is large; Q-Q plot shows minimal deviation from diagonal. |

*Note: Rows show **unstandardized residuals** from the corresponding multivariable regression models (SPSS saved variables: $FEV_1$_zRES_5 = residuals from the $FEV_1$_z model; **FVC_zRES_6** = residuals from the FVC_z model; $FEV_1$/FVC_zRES_7 = residuals from the $FEV_1$/FVC_z model).*

*Abbreviations: K–S = Kolmogorov–Smirnov test; S–W = Shapiro–Wilk test.*

***Normality decision rules (α = 0.01): Accepted** = both tests p ≥ 0.01 and Q–Q plot shows no material deviation; **Partially rejected** = tests disagree (one p ≥ 0.01, one p < 0.01) and only mild tail departure on visual inspection—treated as approximately normal for regression at n = 101; **Rejected** = p < 0.01 on both tests and/or clear departure from normality on Q–Q/detrended plots (interpret model cautiously).*

**of the findings despite the focused sample size**. These findings validate the robustness of the parametric tests used in this study.

Overall, these results demonstrate that **chronic decadal $PM_{2.5}$ exposure is strongly and independently associated with reduced lung volumes** ($FEV_1$_z and FVC_z), while no such impact was observed for obstructive patterns. This enhances the conclusion that the observed lung function deficits are linked to the long-term environmental contrast inherent in this natural experiment.

## Discussion

### Principal findings and comparison with previous studies

In this natural experiment employing a quasi-longitudinal approach and focusing on older adults residing in two urban areas with substantially different ambient PM2.5 concentrations, we found that long-term exposure to higher PM2.5 levels was significantly associated with reduced lung function, specifically reflected by lower $FEV_1$_z and FVC_z scores. This association remained consistent and robust after rigorous adjustment for demographic, socioeconomic, and anthropometric covariates, including Body Mass Index (BMI). The persistence of these findings after controlling for BMI—a known physiological determinant of lung capacity—suggests that the observed decline is independently driven by particulate exposure rather than variations in body composition.

A key methodological strength of this study is the requirement that participants had resided at their current address for at least ten continuous years. This decadal residency threshold provides a quasi-longitudinal perspective that represents a substantial advance over most air pollution epidemiological studies, which often rely on current residence or short-term exposure windows of one to three years. By enforcing this criterion, we minimized exposure misclassification and migration-related bias, thereby strengthening causal inference. Ensuring that prolonged exposure clearly preceded lung function assessment establishes temporal precedence and reduces the risk of reverse causality inherent in standard cross-sectional designs.

Notably, no statistically significant difference was observed in $FEV_1$/FVC_z scores between exposure groups. This pattern—characterized by concurrent reductions in $FEV_1$_z and FVC_z with a preserved ratio—indicates a **non-obstructive, restrictive-pattern spirometry** rather than airflow obstruction. The GOLD 2023 report emphasizes that clinically meaningful lung function impairment may exist in the absence of overt airflow obstruction, particularly among individuals exposed to chronic environmental risk factors such as air pollution [37]. Within this framework, our findings

suggest that the dominant physiological impact of long-term PM2.5 exposure in this older adults manifests primarily as reduced lung volumes, consistent with impaired lung expansion or accelerated pulmonary aging. **The observed effect sizes for (FEV$_1$_z) and (FVC_z) indicate a substantial clinical impact, corresponding to large Cohen's values of 1.21 and 1.02, respectively. In the context of GLI-2012 z-scores, a reduction of more than 1 Standard Deviation (SD) signifies that a significant portion of the exposed population has shifted downward by approximately 35–40 percentile points (e.g., from the median/50th percentile to below the 15th percentile). This framing highlights that the effect is not merely statistically significant but clinically profound at the population level.**

The absence of airflow obstruction is consistent with our study design, which applied stringent exclusion criteria to include only "apparently healthy" older adults, effectively excluding individuals with known chronic respiratory disease, asthma, or significant smoking histories. By isolating the effects of environmental exposure in a clinically asymptomatic cohort, our results demonstrate that substantial physiological impairment can occur before the onset of respiratory symptoms. Accordingly, the lower FEV$_1$_z and FVC_z scores observed in the high-exposure group represent a **subclinical decline in lung capacity** that would likely remain undetected using symptom-based screening or conventional diagnostic thresholds alone. From a clinical standpoint, this 'silent' reduction in lung reserve is critical; while these individuals are currently asymptomatic, their diminished physiological buffer makes them disproportionately vulnerable to acute respiratory failure during minor infections or peak pollution events—a major contributing factor to avoidable emergency hospitalizations.

While clinical practice often relies on fixed spirometric cut-offs for diagnostic convenience, the GOLD 2023 report recognizes the limitations of such approaches in older populations and acknowledges the value of lower-limit-of-normal–based methods for identifying early impairment [37]. **Our study adopts an even more sensitive approach in this regard; as demonstrated by Torén et al. (2020), individuals with ratios between 0.70 and 0.80 already exhibit a significantly higher prevalence of respiratory symptoms, such as dyspnoea (shortness of breath), compared to those with ratios above 0.80. This suggests that the optimal threshold for respiratory health is likely higher, around 0.80–0.85—than the conventional 0.70 cut-off** [38]. In line with this guidance, our use of GLI-2012 z-scores allowed for age-, sex-, height-, and ethnicity-adjusted assessment of lung function, minimizing age-related misclassification and improving sensitivity to early, non-obstructive decline. This approach underscores the importance of volume-based spirometric measures when evaluating the long-term respiratory effects of environmental exposures, particularly in aging populations.

Our findings align with a large body of literature demonstrating the detrimental effects of particulate air pollution on lung function. **However, while massive cohort studies in Europe and North America often struggle with exposure homogeneity or residential mobility [39–41], our natural experiment design isolates the impact of extreme environmental contrast with unprecedented precision** [14–17,40]. Previous research has often found stronger associations for FEV$_1$_z and FVC_z compared with the FEV$_1$/FVC_z ratio, supporting the idea that chronic exposure may contribute more to restrictive-like patterns or reduced lung expansion capacity than to airway obstruction [42]. Our findings contribute additional evidence from Jakarta, one of the most polluted megacities in the world, where older adults may experience particularly high cumulative exposure over the life course.

## Strengths and limitations

This study has several methodological strengths. It focuses on an understudied yet vulnerable population—older adults living in an exceptionally polluted urban environment. Its methodological novelty lies in the application of a strict decade-long residency requirement within a rigorously screened cohort, enabling a cleaner isolation of pollution-related lung function decline than has been achieved in prior global assessments. Unlike conventional environmental studies that treat exposure as a short-term snapshot [43,44], our approach recognizes the aging lung as a product of cumulative environmental insults. By employing a natural experiment design [17–20], this study exploited substantial environmental contrast between two geographically distinct areas, an approach widely regarded as a quasi-gold standard for observational research. This design was intentionally chosen to address the ethical infeasibility of randomized controlled trials in

environmental exposure research, while providing a quasi-experimental framework that strengthens the interpretability of exposure–outcome associations [25–27]. The robustness of the findings is further supported by a post-hoc power analysis indicating that with n = 101 and the observed large effect sizes, the study achieved a statistical power of >0.95 at α = 0.01.

Standardized lung function assessments were conducted in accordance with ATS/ERS guidelines [36], ensuring methodological consistency across measurement conditions. Lung function was evaluated using GLI-2012 z-scores derived from Southeast Asian reference equations, allowing for age-, sex-, height-, and ethnicity-adjusted comparisons and minimizing bias related to ethnic physiological variation. Adherence to GLI-2012 standards and data transparency responds to long-standing calls for methodological consistency and open data practices in respiratory cohort research [41]. Rigorous restriction, frequency matching, and multivariable regression adjustment were employed to control confounding, and residual-based diagnostic tests, **supplemented by Q-Q plots and sensitivity analyses (Cook's Distance < 0.15)**, confirmed the normality, stability and validity of our parametric estimates.

Several limitations should be acknowledged. **Exposure classification was assigned at the area level (Kedoya vs. Pangalengan) and therefore represents an ecological proxy for long-term $PM_{2.5}$ exposure. This approach did not incorporate individual-level personal exposure monitoring, cumulative dose modeling, indoor air quality assessment, or time–activity pattern analysis. Consequently, within-area variability in pollution exposure was not captured, and some degree of residual exposure misclassification cannot be completely excluded. Because exposure was assigned ecologically at the area level, individual heterogeneity in long-term exposure may not be fully represented, and historical exposure trajectories at the individual level were not modeled. These constraints are inherent to ecological exposure assignment and should be considered when interpreting the estimated magnitude of the pollution effect. However, this limitation does not undermine the natural experiment framework, which relies primarily on environmental contrast between populations rather than precise individual-level exposure quantification.**

While the use of two study areas with contrasting exposure profiles enhanced internal validity by maximizing exposure contrast and minimizing contextual heterogeneity, it necessarily constrained exposure variability [45–48]. However, unlike large longitudinal cohorts that may suffer from exposure homogeneity and bias due to unmeasured residential mobility [39–41], our natural experiment design provided a necessary baseline comparison to disentangle pollution-induced deficits from biological aging [14–17,40]. This trade-off between internal validity and exposure variability reflects a deliberate methodological choice to prioritize causal inference. As articulated in the causal uncertainty principle, population restriction can strengthen causal identification at the expense of evidential breadth [49]. **Furthermore, the external validity of these findings is constrained by our strict inclusion criteria. By focusing on a 'highly selected' cohort of non-smoking, non-obese older adults without significant comorbidities, we prioritized internal validity to isolate the independent effect of decadal exposure. Consequently, these results may not be generalizable to populations with different clinical profiles, such as active or former smokers, individuals with obesity, or those with multi-morbidities, in whom the physiological response to chronic air pollution may be further exacerbated or modified by existing systemic inflammation.**

### Translational implications for precision public health

Our findings have direct implications for public health policy in rapidly urbanizing Southeast Asian megacities. The evidence of significant lung volume reduction—even in 'apparently healthy' older adults—suggests that current public health advisories based on general population averages may be insufficient for vulnerable age groups. Consequently, this study underscores the urgent need for robust promotive and preventive public health strategies tailored to the unique risks of the urban older adults.

From a **translational perspective**, this study advocates for **Precision Public Health** that moves beyond broad monitoring toward targeted **respiratory screening and primary prevention** [50,51]. **From a promotive standpoint, health**

**authorities should utilize these findings to enhance community-based education, raising awareness among older adults residents and their caregivers** regarding the long-term respiratory hazards of decadal exposure. Specifically, **integrating spirometry z-score assessments into routine older adults community health posts (*Posbindu*) in high-exposure districts, such as Kedoya, could allow for the early detection of subclinical decline** [51]. Furthermore, these results provide actionable evidence for urban planners to prioritize localized **"clean air zones"** and **specialized indoor filtration subsidies for long-term older adults residents** in areas validated as **high-contrast pollution zones, effectively bridging the gap between environmental data and clinical protection** [52]. **Finally, from a clinical advocacy standpoint,** these findings **empower healthcare practitioners to utilize decadal residential history as a critical, independent risk factor for respiratory decline, ensuring that long-term pollution exposure is prioritized in clinical screenings regardless of a patient's smoking status** [53,54].

Critically, this transition toward a data-driven preventive model serves as a vital safeguard for national healthcare sustainability. By prioritizing early detection within the subclinical window, these targeted surveillance strategies directly mitigate the looming economic burden of chronic respiratory care [54]. Shifting the focus from generalized individual precautions, such as mask-wearing, to systematic screenings and structural interventions ensures a more cost-effective allocation of resources [50,52]. Ultimately, such data-driven prevention at the community level (e.g., Posbindu) acts as a high-value investment, potentially reducing long-term national health insurance (BPJS) expenditures for acute exacerbations and chronic respiratory morbidity among the burgeoning urban elderly population.

## Conclusions

**By utilizing a natural experiment design with a strict 10-year residency requirement**, this study demonstrates that long-term exposure to elevated ambient levels is associated with **markedly reduced lung volumes (FEV$_1$ and FVC z-scores)** among older adults. **By exploiting extreme environmental contrasts as an ethical alternative to randomized trials, this framework—supported by ethnicity-adjusted z-scores—strengthens the interpretation that the observed deficits are attributable to pollution rather than geographic or biological variations. The stability of exposure over a decade, combined with robust model diagnostics confirming the normality of residuals, and model stability (Cook's distance < 0.15), strengthens the evidence for a chronic, cumulative impact compared with standard cross-sectional assessments.**

Crucially, these reductions were observed in a **rigorously screened** population of apparently healthy individuals, indicating that chronic particulate exposure primarily contributes to a subclinical, restrictive-pattern lung function deficit characterized by reduced lung volumes rather than airflow obstruction. In line with the GOLD 2023 report, which emphasizes that clinically meaningful lung function impairment may occur before the development of overt obstructive disease—particularly in the context of chronic environmental exposures—our findings highlight the presence of early physiological vulnerability that remains undetected by traditional symptom-based or fixed-ratio diagnostic thresholds.

**From a socio-economic perspective, these findings suggest that long-term urban residency may function as a structural determinant of health, where geography influences patterns of respiratory aging. This underscores the necessity of a targeted public health approach that moves beyond individual-focused advice, such as intermittent mask-wearing, toward systemic interventions.** Utilizing standardized z-score–based spirometry to identify pollution-related lung function decline at a subclinical stage enables **earlier risk stratification** and **preventive intervention** among vulnerable aging populations. Furthermore, **institutionalizing this framework within national health systems (such as Indonesia's BPJS)** may offer a strategic pathway to **mitigate the future economic burden of emergency hospitalizations** and **chronic respiratory morbidity.**

**Ultimately, this research provides the evidentiary foundation for a dual-track strategy in public health: robust health promotion to increase community awareness of long-term hazards, and precise primary prevention**

**through routine respiratory screening in high-pollution corridors.** These findings provide **actionable translational evidence** for shifting public health strategies from broad, population-wide advisories toward **targeted, data-driven interventions**.

Beyond general air-quality regulations, this study advocates for the integration of decadal environmental risk assessments into routine geriatric clinical care. **The translational value of this research** lies in **its capacity to bridge the gap between environmental monitoring** and **clinical protection**, **supporting a framework that prioritizes high-resolution screening for long-term residents in validated pollution hotspots.** While our **quasi-longitudinal approach** provides **robust evidence of the chronic impact of decadal exposure**, future prospective longitudinal studies could further illuminate the real-time rate of lung function decline and the potential reversibility of these subclinical deficits in populations transitioning between different pollution trajectories.

### Recommendations

Based on the evidence of subclinical lung function deficits associated with long-term $PM_{2.5}$ exposure among biologically and environmentally vulnerable older adults, this study supports **risk-stratified prevention and surveillance** that moves beyond population-wide recommendations toward risk-stratified prevention and surveillance.

**At the policy and macro-economic level**, our findings underscore the need for **precision regulation** in high-burden urban corridors. Rather than uniform air quality interventions, municipalities should prioritize **high-resolution $PM_{2.5}$ monitoring** in densely populated areas with long-standing exposure, particularly where older adults reside. These data can inform the designation of **respiratory protection priority zones, in which environmental mitigation efforts**—such as **stricter localized emission controls**, **traffic regulation, and urban greening**—are preferentially implemented to protect populations with the highest cumulative exposure and biological vulnerability. **Crucially, we recommend that health policymakers integrate these findings into national health insurance frameworks (e.g., BPJS) by reallocating resources toward preventive respiratory screenings within these designated zones. Investing in early detection at the community level acts as a fiscal safeguard, potentially reducing the long-term economic burden of emergency hospitalizations and chronic oxygen dependency.**

At the **clinical and public health service perspective**, the observed reductions in $FEV_1\_z$ and $FVC\_z$ among asymptomatic older adults highlight the **importance of early, risk-based detection strategies. Routine spirometric screening using standardized GLI-2012 z-scores** should be **integrated** into **primary care and community health services** for older adults living in high-pollution areas, particularly those with long-term residential stability. This targeted surveillance approach enables the identification of subclinical lung function decline before overt disease develops, allowing **preventive interventions** to be initiated at a reversible stage. For individuals identified as highly vulnerable—based on cumulative exposure and genetic susceptibility—**enhanced follow-up** and **individualized respiratory health monitoring** may be warranted.

At **the individual and household level**, **precision prevention** should focus on **environmental modification** rather than reliance on behavioral restriction alone. Interventions such as **improving indoor air quality through ventilation optimization**, **the use of air filtration systems**, and **tailored activity planning** during periods of elevated pollution represent more sustainable strategies for protecting the aging lung. While **personal protective measures**, including the use of well-fitted respirators (e.g., N95 or KN95 masks), may serve as **supplementary short-term mitigation** during peak pollution events, they **should be viewed as part of a broader**, **integrated prevention strategy rather than a stand-alone solution.**

Finally, this study provides a **foundation for future research** aimed at strengthening mechanistic and causal understanding. **Longitudinal follow-up** with repeated spirometric assessments would allow estimation of individual rates of lung function decline, while the incorporation of inflammatory and oxidative stress **biomarkers** could further elucidate biological pathways underlying pollution-related lung impairment. Additionally, **intervention studies** evaluating indoor air

quality improvements or community-level emission reductions would be valuable to determine whether targeted environmental modifications can **halt or potentially reverse subclinical lung function loss** in high-risk older adults.

## Supporting information

**S1 File. Supplementary materials.** This file contains the completed STROBE checklist, S1 Table, S2 Table, and S1 Fig. (PDF)

## Author contributions

**Conceptualization:** Hari Krismanuel.

**Data curation:** Purnamawati Tjhin.

**Formal analysis:** Hari Krismanuel.

**Investigation:** Hari Krismanuel.

**Methodology:** Hari Krismanuel.

**Visualization:** Purnamawati Tjhin.

**Writing – original draft:** Hari Krismanuel.

**Writing – review & editing:** Hari Krismanuel.

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
