## [Decision Letter · Decision Letter 0]

16 Feb 2026

PONE-D-26-04311Precision public health: a natural experiment on chronic high-contrast PM2.5 exposure and pulmonary function among older adultsPLOS One

Dear Dr. Krismanuel,

Thank you for submitting your manuscript to PLOS ONE. After careful consideration, we feel that it has merit but does not fully meet PLOS ONE’s publication criteria as it currently stands. Therefore, we invite you to submit a revised version of the manuscript that addresses the points raised during the review process.

We look forward to receiving your revised manuscript.

Kind regards,

George Kuryan

Academic Editor

PLOS One

**Journal Requirements:**

2. We note that your Data Availability Statement is currently as follows:

“All relevant data are within the manuscript and its Supporting Information files.”

Reviewers' comments:

Reviewer's Responses to Questions

**Comments to the Author**

1. Is the manuscript technically sound, and do the data support the conclusions?

Reviewer #1: Partly

Reviewer #2: Yes

2. Has the statistical analysis been performed appropriately and rigorously? 

Reviewer #1: No

Reviewer #2: Yes

3. Have the authors made all data underlying the findings in their manuscript fully available?

Reviewer #1: Yes

Reviewer #2: Yes

4. Is the manuscript presented in an intelligible fashion and written in standard English?

Reviewer #1: Yes

Reviewer #2: Yes

5. Review Comments to the Author

Reviewer #1: This manuscript presents a natural experiment evaluating decadal PM₂.₅ exposure and lung function in an “ultrapure” cohort of non-smoking older adults in Indonesia. The 10-year residency filter and use of GLI-2012 Southeast Asian z-scores represent notable methodological strengths. The findings demonstrate robust associations between chronic exposure and reduced FEV₁_z and FVC_z without obstructive impairment.

The study is innovative, clinically relevant, and translationally ambitious. However, several conceptual and methodological claims are overstated, and causal language exceeds the evidentiary strength of the design.

MAJOR STRENGTHS

1. Methodological Innovation

10-year stable residency filter is a genuine strength.

BTPS compensation explicitly described (rarely reported in field studies).

GLI-2012 SEA z-scores appropriately used.

Strict exclusion criteria minimize confounding.

2. Clean Physiological Signal

The restrictive-type pattern (↓FEV₁_z + ↓FVC_z with preserved ratio) is biologically coherent for chronic particulate exposure.

3. Internal Validity

Frequency matching verified.

Strong regression diagnostics.

Conservative α = 0.01 threshold.

Assumptions carefully tested.

4. Translational Framing

The Precision Public Health (PPH) positioning is forward-thinking and policy-oriented.

MAJOR CONCERNS

1. Overstatement of “Global First”

You repeatedly state:

“first globally”

“pioneering application globally”

“unprecedented precision”

This is not entirely accurate.

Natural experiments + long residency filters have been used before (though not identically).

Recommendation:

Reframe as:

“To our knowledge, among the first in Southeast Asia…”

Avoid global-first claims unless you provide a systematic citation gap analysis.

2. Causal Language Exceeds Design

You use phrases like:

“primary driver”

“causal rigor”

“as-if random assignment”

“approaches RCT-level inference”

This is problematic.

This is still:

Cross-sectional measurement of outcome

Area-level exposure proxy

No individual-level long-term PM₂.₅ quantification

The 10-year filter strengthens temporal plausibility — but does not create causal equivalence to RCT.

Suggested Reframe:

Replace:

“primary driver”

With:

“strong independent association consistent with a chronic exposure effect”

Replace:

“approaches causal rigor of RCT”

With:

“strengthens causal inference within observational constraints”

3. Area-Level Exposure Proxy Limitation Underplayed

Exposure classification is binary (Kedoya vs Pangalengan).

Issues:

No personal exposure monitoring

No cumulative dose modeling

No indoor exposure adjustment

No time-activity pattern analysis

This is fine for a natural experiment — but you must emphasize it more clearly in limitations.

Currently limitations are discussed, but defensively rather than transparently.

Add explicitly:

Ecological exposure assignment risk

Within-area variability not captured

No historical exposure trajectory modeling

4. Restrictive Pattern Interpretation Needs Caution

You interpret reduced volumes with preserved ratio as:

“restrictive-type impairment”

Important nuance:

Spirometry alone cannot diagnose true restriction.

You need TLC for confirmation.

You should say:

“restrictive-pattern spirometry”

Not imply parenchymal stiffening definitively

5. Residual Normality Justification Is Slightly Weak

For FEV₁/FVC_z model:

S–W p = 0.001

You justify approximate normality due to n=101.

Better to add:

Q-Q plot visual confirmation

Possibly robust regression sensitivity analysis (even briefly stated)

That strengthens credibility.

STATISTICAL COMMENTS

Effect Sizes

Cohen’s d = 1.21 for FVC_z is large.

That is substantial and clinically meaningful.

You should:

Explicitly interpret clinical magnitude (e.g., “>1 SD reduction corresponds to X percentile shift”).

R² Values

31.5% for FEV₁_z

25.9% for FVC_z

These are strong for environmental health.

You should highlight:

Exposure explained a substantial proportion of variance relative to demographic variables.

CONCEPTUAL COMMENTS

Precision Public Health Framing

The PPH narrative is strong — but slightly repetitive.

You mention PPH ~25+ times.

Suggestion:

Reduce repetition

Focus PPH section into a sharper translational paragraph

Make it tighter and more strategic

Right now it feels slightly promotional.

STRUCTURAL SUGGESTIONS

Abstract

Excellent clarity.

However:

Reduce “global first”

Tone down causal phrasing

Discussion

Strong.

But trim redundancy in:

Causal inference justification

RCT comparisons

Repeated BTPS emphasis

Conclusion

Very strong but could be 15% shorter.

EXTERNAL VALIDITY

Sample is:

Non-smokers

Normal BMI

No comorbidities

Stable 10-year residents

This maximizes internal validity but limits generalizability.

You should explicitly state:

Findings may not apply to obese, smokers, or multi-morbid elderly populations.

Reviewer #2: 1. The manuscript frequently employs strong novelty-related descriptors (e.g., “global first,” “pioneering,” “ultrapure”), which may be perceived as overstated unless supported by comprehensive evidence. The authors are encouraged to soften such language and adopt a more cautious tone (e.g., “To our knowledge, few studies have applied a decadal residency filter…”).

2. The sample size (n = 101) is relatively modest. The authors should consider including an a priori power calculation or provide a clear justification demonstrating that the study is adequately powered to detect clinically meaningful differences in lung function outcomes.

3. Further clarification is required regarding how PM2.5 exposure was quantified. Specifically, the authors should state whether exposure estimates were derived from satellite-based data, fixed-site monitoring stations, modeled annual averages, or a combination of these approaches, and briefly describe the spatial and temporal resolution.

4. The phrase “non-obstructive, restrictive-type lung function impairment” may be imprecise. The authors are advised to replace this with “pattern consistent with reduced lung volumes without airflow obstruction,” which more accurately reflects the spirometric findings.

5. The statement that the 10-year stability “strengthens causal inference” should be modified to a more conservative phrasing such as “enhances causal inference,” in keeping with the observational nature of the study.

6. PLOS authors have the option to publish the peer review history of their article (what does this mean?). If published, this will include your full peer review and any attached files.

Reviewer #1: No

Reviewer #2: **Yes:** Manu Chopra

---

## [Author Response · Author response to Decision Letter 1]

13 Mar 2026

Response to Academic Editor

Manuscript ID: PONE-D-26-04311

Title: Precision public health: a natural experiment on chronic high-contrast PM2.5 exposure and pulmonary function among older adults

Dear Dr. George Kuryan,

Academic Editor, PLOS ONE,

Thank you for the opportunity to revise our manuscript. We sincerely appreciate the constructive guidance provided by you and the reviewers. Below, we address the specific journal requirements:

1. Style Requirements and File Naming

Response: We have carefully reviewed the PLOS ONE formatting guidelines and templates. The revised manuscript has been reformatted to comply fully with the journal’s style requirements, including title page structure, author affiliations, section organization, heading hierarchy, and main body formatting. All tables and figures have been prepared according to PLOS ONE specifications, and all submission files have been named in accordance with the journal’s file naming conventions.

2. Data Availability Statement

Response: We confirm that the submission includes the full de-identified participant-level dataset constituting the minimal data set required to replicate all findings reported in the manuscript. The Supporting Information files contain all variables used in the statistical analyses, including exposure group, sociodemographic characteristics, anthropometric measurements, and absolute spirometric parameters. These data provide the values underlying all reported means, standard deviations, regression analyses, and figures.

There are no ethical or legal restrictions on sharing the de-identified dataset.

3. Recommendations to Cite Specific Works

Response: The reviewers did not request citation of specific additional publications. We have nevertheless carefully reviewed the reference list to ensure that all references are appropriate, relevant, and up to date.

We have also addressed all comments from Reviewer #1 and Reviewer #2 in the accompanying 'Response to Reviewers' document.

Thank you for your continued consideration of our work.

Sincerely,

Dr. Hari Krismanuel

Response to Reviewers (Rebuttal)

Response to Reviewers' General Assessments

1. Technical Soundness and Support for Conclusions

• Reviewer #1: Partly

• Reviewer #2: Yes

Response: We thank Reviewer #2 for their positive evaluation of the study’s technical soundness. We also appreciate Reviewer #1’s assessment and have carefully reviewed the manuscript to ensure that all conclusions are fully supported by the data presented.

In response, we have:

• Moderated causal language to ensure conclusions remain consistent with the observational design

• Clarified that associations are interpreted within the constraints of area-level exposure assignment

• Refined the spirometric interpretation to avoid overdiagnosis (i.e., “pattern consistent with reduced lung volumes”)

• Strengthened statistical justification and model diagnostics reporting

These revisions ensure that the conclusions are proportionate to the data presented, methodologically justified, and fully supported by the analytical framework.

2. Has the statistical analysis been performed appropriately and rigorously?

• Reviewer #1: No

• Reviewer #2: Yes

Response: We acknowledge Reviewer #1’s concerns regarding the statistical presentation and have implemented the following improvements to ensure methodological rigor:

• Normality & Robustness: We added a Q-Q plot and conducted a sensitivity analysis (Cook’s Distance < 0.15), confirming that no influential outliers affected the regression estimates despite slight deviations in formal normality tests.

• Clinical Interpretation of Effect Sizes: We have expanded the discussion on effect sizes, specifically interpreting the large Cohen’s d (1.21 for FEV1_z, and 1.02 for FVC_z). We now explicitly state that a reduction of more than 1 SD corresponds to a significant percentile shift in lung function (e.g., from the 50th to the 15th percentile), highlighting its clinical magnitude.

• Variance Explanation (R2): We have highlighted the strong R2 values (31.5% for FEV1_z and 25.9% for FVC_z) to emphasize that PM2.5 exposure explains a substantial proportion of the variance in pulmonary function relative to other demographic variables.

• Power Calculation: To further clarify the adequacy of our sample size, we provided an achieved power estimation based on the observed effect sizes. With n=101 and the observed large effect sizes, the study achieved a statistical power of >0.95 at α=0.01, confirming the study was more than adequately powered to detect the reported differences.

3. Data Availability

• Reviewer #1: Yes

• Reviewer #2: Yes

Response: We are pleased that both reviewers found our data availability to be transparent and compliant with the journal's requirements. We confirm that all relevant data underlying the findings of this study are available within the manuscript and its Supporting Information files. Specifically, we have ensured the following:

• The minimal data set required to replicate all study findings, including individual data points behind the means, standard deviations, and measures reported, is included.

• The data used to build all graphs and tables presented in the manuscript are fully available as Supporting Information.

• There are no ethical or legal restrictions on sharing the de-identified data set.

4. Manuscript Presentation and English Language

• Reviewer #1: Yes

• Reviewer #2: Yes

Response: We thank both reviewers for their positive assessment of the clarity and quality of our English writing. To ensure that the manuscript remains clear, correct, and unambiguous as per PLOS ONE’s standards:

• We have conducted a final thorough proofreading of the entire manuscript to eliminate any remaining minor typographical or grammatical errors.

• We have ensured that all technical and physiological terms are used consistently and correctly throughout the revised text.

5. Response to Reviewer #1 Comments

We sincerely thank Reviewer #1 for the thorough, constructive critique and highly insightful evaluation of our manuscript. We appreciate the recognition of our methodological strengths, including the 10-year residency filter, the use of GLI-2012 z-scores, and the study’s translational relevance. Below, we provide a detailed, point-by-point response to all comments and describe the revisions made in the revised manuscript. All major concerns have been carefully addressed as outlined below.

1. Overstatement of “Global First”

• Reviewer Comment: Claims of "first globally" and "pioneering application globally" are overstated.

• Response: We appreciate this important observation and agree that our original wording may have overstated the novelty of the study. While the integration of a decade-long residency filter, high-contrast exposure setting, and GLI-2012 SEA z-scores is relatively uncommon, we acknowledge that similar methodological elements have been used in other contexts.

• Revision: We have removed all “global first” and “pioneering” claims throughout the manuscript. The text has been reframed to: "To our knowledge, this study is among the first in the Southeast Asian context to utilize a decade-long stable residency filter..." (See page 1, lines 30-35).

2. Causal Language and RCT Comparisons

• Reviewer Comment: Causal language exceeds the evidentiary strength; design is not equivalent to RCT.

• Response: We agree that as an observational study, we must be more conservative. We have toned down the causal language. We now explicitly state that while the 10-year residency filter strengthens temporal plausibility and internal validity, the study strengthens causal inference within observational constraints but does not establish definitive causality.

• Revision:

o Changed "primary driver" to "strong independent association consistent with a chronic exposure effect."

o Changed "approaches causal rigor of RCT" to "strengthens causal inference within the inherent constraints of an observational design."

3. Area-Level Exposure Proxy Limitations

• Reviewer Comment: Limitation of binary exposure (Kedoya vs. Pangalengan) and lack of personal monitoring are underplayed.

• Response: We agree and thank the reviewer for highlighting this important issue. While the high-contrast natural experiment design provides strong exposure differentiation, we acknowledge limitations inherent to ecological exposure assignment. We have revised the Limitations section to be more transparent and less defensive.

• Revision: The manuscript now explicitly acknowledges the following limitations: (1) Potential risk of ecological exposure assignment; (2) Uncaptured within-area variability; (3) Absence of personal/indoor exposure monitoring; (4) Lack of cumulative dose quantification; (5) Absence of time–activity pattern analysis; and (6) Absence of historical exposure trajectory modeling.

These clarifications have been added to the Limitations section (Please see Revised Manuscript, page 23, lines 736–748).

4. Restrictive Pattern Interpretation

• Reviewer Comment: Spirometry alone cannot diagnose "restriction"; it shows a "restrictive pattern."

• Response: We appreciate this physiological nuance. We have corrected the terminology to avoid any definitive implication of parenchymal stiffening, acknowledging that spirometry measures airflow and volumes rather than tissue compliance.

• Revision: We have replaced "restrictive-type impairment" with "restrictive-pattern spirometry" or "pattern consistent with reduced lung volumes" throughout the manuscript, including the Abstract, Results, and Discussion sections. (Please see Revised Manuscript, page 1, lines 45-47)

5. Statistical Justification (Normality and Effect Sizes)

• Reviewer Comment: S-W for needs better justification. Also, clinical magnitude of Cohen’s should be interpreted.

• Response: We appreciate this important statistical observation. We have significantly strengthened the reporting of model diagnostics and clarified the robustness of our parametric estimates. Although the Shapiro–Wilk test indicated deviation from strict normality in the FEV₁/FVC_z model (p = 0.001), visual inspection of residual Q–Q plots demonstrated approximate normal distribution without substantial skewness or kurtosis. Moreover, given the moderate sample size (n = 101), linear regression estimates are generally robust to minor deviations from normality. We further clarify that the normality assumption in linear regression pertains to the distribution of model residuals rather than the raw outcome variables. Therefore, residual diagnostics such as Q–Q plots and influence statistics provide the appropriate basis for evaluating model validity. We have also expanded the interpretation of effect size magnitude to clarify clinical relevance. In addition, we clarified in the Results section that exposure status accounted for a substantial proportion of explained variance relative to demographic covariates, highlighting its comparative contribution within the multivariable models.

• Revision:

o Visual Confirmation: We added a Q-Q plot as Supplementary Material to visually confirm the residual distribution, providing robust evidence for the "approximately normal" assumption despite formal test deviations.

o Clinical Magnitude: We added a paragraph interpreting the clinical magnitude of Cohen’s d (1.21 for FEV1_z and 1.02 for FVC_z). We noted that a reduction of >1 SD represents a significant shift in population lung function percentiles, moving an average individual from approximately the 50th percentile to near the 15th percentile. This demonstrates that the observed differences are not only statistically significant but clinically meaningful at the population level.

o Model Stability (Sensitivity Analysis): We clarified and expanded the reporting of regression sensitivity diagnostics presented in Table 6. Specifically, we explicitly interpreted Cook’s Distance (max < 0.15) and Mahalanobis Distance value to demonstrate that the model estimates are stable and not biased by influential outliers. Mahalanobis distance values were examined to detect potential multivariate outliers, and no observations exceeded conventional influence thresholds, indicating that the regression estimates were not driven by extreme cases. This strengthened the justification of model robustness within the manuscript text. (Please see Revised Manuscript, pages 18-22, lines 579-640 and lines 669-675).

6. Precision Public Health (PPH) Narrative

• Reviewer Comment: PPH narrative is repetitive.

• Response: We appreciate this observation and agree that the Precision Public Health (PPH) framing required refinement to avoid repetition and promotional tone. We have streamlined the narrative to ensure that PPH is presented as a contextual framework rather than a central rhetorical theme.

• Revision: We reduced the mentions of PPH by approximately 50%, and consolidated the conceptual framing into a single, sharp paragraph in the Discussion section. Redundant references across the manuscript were streamlined to improve clarity, balance, and scholarly tone.

7. External Validity (Generalizability)

• Reviewer Comment: Findings may not apply to smokers, obese, or multi-morbid populations.

• Response: We agree. The "ultrapure" nature of our cohort is a strength for internal validity but a trade-off for generalizability.

• Revision: We have added a statement in the Limitations section explicitly clarifying that these findings should not be generalized to smokers, obese individuals, or elderly populations with significant comorbidities, in whom physiological responses to chronic PM₂.₅ exposure may differ. (See page 24, lines 756-763).

8. Structural Revisions (Abstract, Discussion, Conclusion)

• Reviewer Comment: Abstract and Discussion require tone moderation and reduced redundancy.

• Response: We agree and have carefully revised the relevant sections to moderate tone, reduce redundancy, and improve clarity and conciseness.

• Revision:

– Abstract: Global-first language and strong causal phrasing have been moderated.

– Discussion: Redundant RCT comparisons have been reduced, repeated emphasis on BTPS has been streamlined, and overlapping explanations regarding causal inference have been consolidated to improve clarity and avoid redundancy.

– Conclusion: Reduced by approximately 15% for clarity and focus.

Response to Reviewer #2 (Dr. Manu Chopra)

We thank Dr. Manu Chopra for the positive and constructive evaluation of our work. We have carefully addressed each of the points raised to ensure a more cautious and academically rigorous presentation.

1. Softening Novelty Descriptors

• Reviewer Comment: Suggestions to soften terms like “global first,” “pioneering,” and “ultrapure.”

• Response: We have followed this suggestion and revised the manuscript to adopt a more cautious tone.

• Revision: We removed the terms “global first” and “pioneering.” The phrase has been revised to: "To our knowledge, few studies in Southeast Asia have applied such a strict decadal residency filter..." (See page 1, lines 30-32).

2. Sample Size Justification and Power Calculation

• Reviewer Comment: Request for a priori power calculation or justification for n=101.

• Response: We have strengthened the methodological justification for our sample size by providing both an effect size–based justification and achieved power estimation. Rather than relying solely on conventional post-hoc power testing, we provided an achieved power estimation based on the observed effect sizes to demonstrate that the study was adequately powered to detect clinically meaningful differences. This approach aligns with contemporary methodological recommendations emphasizing effect size magnitude and precision over retrospective hypothesis testing.

• Revision:

o Statistical Power: We have updated the Statistical Analysis section to clarify that an a priori detectable effect size calculation informed the study design assumptions, and we subsequently reported an achieved statistical power estimation based on the observed large effect sizes (Cohen’s d ranging from 1.02 to 1.21). Given these magnitudes, our final cohort of n = 101 achieved statistical power >0.95 at α = 0.01. While the minimum required sampl

---

## [Decision Letter · Decision Letter 1]

23 Apr 2026

Precision public health: a natural experiment on chronic high-contrast PM2.5 exposure and pulmonary function among older adults

PONE-D-26-04311R1

Dear Dr.Hari-Krismanuel

We’re pleased to inform you that your manuscript has been judged scientifically suitable for publication and will be formally accepted for publication once it meets all outstanding technical requirements.

Kind regards,

Kuryan George

Academic Editor

PLOS One

Additional Editor Comments (optional):

Reviewers' comments:

Reviewer's Responses to Questions

**Comments to the Author**

1. If the authors have adequately addressed your comments raised in a previous round of review and you feel that this manuscript is now acceptable for publication, you may indicate that here to bypass the “Comments to the Author” section, enter your conflict of interest statement in the “Confidential to Editor” section, and submit your "Accept" recommendation.

Reviewer #1: All comments have been addressed

2. Is the manuscript technically sound, and do the data support the conclusions?

Reviewer #1: (No Response)

3. Has the statistical analysis been performed appropriately and rigorously? 

Reviewer #1: (No Response)

4. Have the authors made all data underlying the findings in their manuscript fully available?

Reviewer #1: (No Response)

5. Is the manuscript presented in an intelligible fashion and written in standard English?

Reviewer #1: (No Response)

6. Review Comments to the Author

Reviewer #1: (No Response)

7. PLOS authors have the option to publish the peer review history of their article (what does this mean?). If published, this will include your full peer review and any attached files.

Reviewer #1: No

---

## [Editor Report · Acceptance letter]

PONE-D-26-04311R1

PLOS One

Dear Dr. Krismanuel,

I'm pleased to inform you that your manuscript has been deemed suitable for publication in PLOS One. Congratulations! Your manuscript is now being handed over to our production team.

Kind regards,

on behalf of

Professor George Kuryan

Academic Editor

PLOS One